# Differences in Pulmonary Function Improvement after Once-Daily LABA/LAMA Fixed-Dose Combinations in Patients with COPD

**DOI:** 10.3390/jcm11237165

**Published:** 2022-12-01

**Authors:** Wei-Chun Huang, Chih-Yu Chen, Wei-Chih Liao, Biing-Ru Wu, Wei-Chun Chen, Chih-Yen Tu, Chia-Hung Chen, Wen-Chien Cheng

**Affiliations:** 1Division of Pulmonary and Critical Care, Department of Internal Medicine, China Medical University Hospital, No. 2, Yude Road, North District, Taichung 40402, Taiwan; 2School of Medicine, College of Medicine, China Medical University, Taichung 406, Taiwan; 3Ph.D. Program in Translational Medicine, National Chung Hsing University, Taichung 402, Taiwan; 4Rong Hsing Research Center for Translational Medicine, National Chung Hsing University, Taichung 402, Taiwan

**Keywords:** COPD, chronic obstructive pulmonary disease, UMEC/VIL, umeclidinium/vilanterol, IND/GLY, indacaterol/glycopyrronium, TIO/OLO, tiotropium/olodaterol

## Abstract

This real-world study evaluated the efficacy of once-daily long-acting β2-agonist (LABA)/long-acting muscarinic antagonist (LAMA) fixed-dose combinations (FDCs) for improving lung function in patients with chronic obstructive pulmonary disease (COPD). Patients with COPD who were treated with once-daily LABA/LAMA FDCs for 12 months were included. We evaluated their lung function improvement after 12 months of treatment with different LABA/LAMA FDCs. A total of 198 patients with COPD who were treated with once-daily LABA/LAMA FDCs were analyzed. A total of 114 patients were treated with umeclidinium/vilanterol (UMEC/VIL); 34 patients were treated with indacaterol/glycopyrronium (IND/GLY); and 50 patients were treated with tiotropium/olodaterol (TIO/OLO). The forced expiratory volume in 1 s (FEV1) was significantly increased in the patients treated with all three once-daily FDCs (55.2% to 60.9%, *p* = 0.012 for UMEC/VIL, 58.2% to 63.6%, *p* = 0.023 for IND/GLY, and 54.1% to 57.7%, *p* = 0.009 for TIO/OLO). The treatment of COPD patients with TIO/OLO resulted in a significant improvement in both forced vital capacity (FVC%) (71.7% to 77.9%, *p* = 0.009) and residual volume (RV%) (180.1% to 152.5%, *p* < 0.01) compared with those treated with UMEC/VIL (FVC%: 75.1% to 81.5%, *p* < 0.001; RV%:173.8% to 165.2%, *p* = 0.231) or IND/GLY (FVC%: 73.9% to 79.3%, *p* = 0.08; RV%:176.8% to 168.3%, *p* = 0.589). Patients treated with UMEC/VIL or TIO/OLO showed significant improvement in FVC. In addition, those receiving TIO/OLO also showed significant improvement in RV reduction.

## 1. Introduction

Inhaled bronchodilators are the main pharmacological therapy for each stage of chronic obstructive pulmonary disease (COPD) [1]. There is a lot of evidence to show that once-daily long-acting β2-agonist (LABA)/long-acting muscarinic antagonist (LAMA) dual bronchodilators are superior for symptom control and lung function improvement compared with monotherapy [2,3,4,5]. Several studies also reported that the early use of LABA/LAMA dual bronchodilators may improve disease stability and prevent further disease deterioration compared to monotherapy, including treatment-naive patients [6,7,8]. As a result, LABA/LAMA dual bronchodilators are recommended as the initial therapy for COPD patients with more symptoms, a high frequency of acute exacerbations (AEs), and poor lung function.

Nowadays, several different once-daily LABA/LAMA fixed-dose combination (FDC) dual bronchodilator regimens are available, including (1) umeclidinium/vilanterol (UMEC/VIL) 55 μg/22 μg via an Anoro Ellipta^TM^; (2) tiotropium/olodaterol (TIO/OLO) 2.5 μg/2.5 μg via a Spiolto Respimat^TM^; and (3) indacaterol/glycopyrronium (IND/GLY) 110 μg/50 μg via an Ultibro Breezhaler^TM^. The comparative efficacy of these different dual bronchodilator combinations has not been widely studied in symptomatic COPD patients. Several indirect comparison studies have tried to compare the efficacy of currently available LABA/LAMA FDC dual bronchodilators for COPD via network meta-analyses [9,10,11]; they revealed that there was no significant difference in lung function improvement among the different medications. A few head-to-head studies revealed that UMEC/VIL regimens had larger increases in the forced expiratory volume in 1 s (FEV1) compared with TIO/OLO [12,13]. However, in another study, there was no significant change in FEV1 after the inhalation of UMEC/VIL compared with IND/GLY [14].

Few real-world studies have compared lung function efficacy in COPD patients administered one of the three once-daily LABA/LAMA FDCs. Cheng et al. conducted a real-word study in Taiwan and reported that TIO/OLO resulted in superior lung function improvement (FEV1 and force vital capacity (FVC)) and reduced AEs, compared with the other regimens (IND/GLY and UMEC/VI) [15]. Muraki et al. reported that there were no significant differences in the COPD Assessment Test (CAT), the modified Medical Research Council (mMRC) dyspnea questionnaire, and spirometry items following the administration of the three FDCs [16]. The efficacy of dual bronchodilators for lung function improvement are variable, and there are many factors that can affect the efficiency of medication in real-world practice, such as adherence to medication, the frequency of administration, the role of the device, and the patients’ understanding and views of the disease and treatment [17,18,19]; these factors cannot be seen or controlled for in randomized controlled trials (RCTs). In addition, in most studies of COPD, only FEV1 has been focused on and used as an outcome measurement. However, the calculation of FEV1 alone has limitations due to the underlying heterogeneous nature of COPD, and it is widely accepted that FEV1 measurement alone does not effectively represent functional impairment [20]. A comprehensive patient evaluation, including multiple parameters, is required for efficient COPD phenotyping [21]. As a result, more parameters of pulmonary function are needed to evaluate the efficacy of the different LAMA/LABA FDCs in COPD patients. This real-world study aimed to determine whether the three different once-daily LAMA/LABA FDCs (i.e., UMEC/VIL, IND/GLY, and TIO/OLO) would produce similar benefits in regular clinic patients without close supervision, as in clinical trial patients, with regard to objectively assessed pulmonary function parameters (FEV1, FVC, and RV) in a heterogeneous population with COPD.

## 2. Methods

### 2.1. Study Patients

This study was conducted retroactively at the Division of Pulmonary and Critical Care Medicine, China Medical University Hospital, Taiwan between December 2014 and September 2020. The inclusion criteria of patients were as follows: (1) over 40 years old, (2) diagnosed with COPD based on clinical symptoms (such as breathing difficulty, cough, sputum production, and wheezing) and spirometry (post-bronchodilator FEV1/FVC < 0.7), and lifetime risk for the development of COPD; (3) patients who were being treated with the same once-daily LAMA/LABA FDCs continuously for at least 52 weeks or 12 months. Patients may have received other therapy such as monotherapy or ICS + LABA before inclusion in the study. Patients were excluded if they were treated with triple therapy, combination therapy (ICS/LABA), or mono therapy (LAMA or LAMA) in the final analysis, or they had insufficient data available for further analysis. The study was approved by the China Medical University Hospital Institutional Review Board (CMUH110-REC1-245), and the need for informed consent was waived due to the observational and retrospective design of the study. The once-daily LABA/LAMA FDC included: (1) UMEC/VIL 55 μg/22 μg via an Anoro Ellipta^TM^; (2) TIO/OLO 2.5 μg/2.5 μg via a Spiolto Respimat^TM^; (3) IND/GLY 110 μg/50 μg via an Ultibro Breezhaler^TM^. In order to improve their health outcomes and healthcare quality, all patients participated in healthcare case management.

### 2.2. Clinical Data Collection and Treatment Assessment

We retrospectively reviewed the data of all patients with COPD who were treated with the same once-daily LAMA/LABA FDCs continuously for at least 1 year between December 2014 and September 2020. The data at the initial treatment with LABA/LAMA FDCs were collected in the first 12 months. The clinical data of the patients, including age, sex, body height (BH), body weight (BW), body mass index (BMI), smoking status, history of asthma, comorbidities, pulmonary function tests (PFTs), CAT score, mMRC score, acute exacerbation in the previous year, and GOLD airflow limitation severity were collected. GOLD airflow limitation was divided into 4 stages based on the patient’s FEV1% predicted: stage 1/mild, FEV1 ≥ 80%; stage 2/moderate, 50–79%; stage 3/severe, 30–49%; stage 4/very severe FEV1 < 30% [22]. After the primary treatment was started, patients underwent regular (every three to six months) follow-ups at our institute. The pulmonary function parameters (FEV1, FVC, and RV) at 12 months of treatment were compared with the pre-LABA/LAMA FDC treatment values. Spirometry was performed before the administration of the daily dose of LABA/LAMA FDCs. FEV1 and FVC were measured through the method of spirometry. RV was determined through body plethysmography, with the usage of Body Box—62J with the Vmax Encore 22D System. The prediction values of FEV1, FVC, and RV were based on the Knudson equations adopted during the 1980s, which were based on the patients’ age, sex, and body height. The values of FEV1% or FVC% were derived from FEV1 or FVC measurements through spirometry divided by the predicted FEV1 or FVC in a population of similar age, sex, and body composition. RV% was defined as the RV measured through body plethysmography divided by the predicted value of the RV.

### 2.3. Statistical Analysis

Continuous variables were presented as median and interquartile range (IQR; 25th and 75th percentiles) or mean with standard deviation (SD). The baseline clinical characteristics of each group included age, sex, body height, body weight, BMI, smoking pack/year, and pulmonary function test parameters. These were analyzed via analysis of variance (ANOVA), followed by the Tukey test for specific comparisons between means. The CAT score and exacerbation history of each group were analyzed via the Kruskal–Wallis test and post hoc Dunn’s multiple comparisons test. Categorical variables were reported as the number of patients and percentages. Differences in categorical variables were examined using the chi-squared test. A paired *t*-test was used to compare the mean FEV1, FEV1%, FVC, FVC%, and RV% before and after the LABA/LAMA FDCs therapy. All tests of significance were two sided, and a *p*-value ≤ 0.05 was considered to indicate a statistically significant difference. All statistical analyses were performed using MedCalc for Windows, version 18.10 (MedCalc Software, Ostend, Belgium).

## 3. Results

### 3.1. Baseline Characteristics

A total of 198 patients with COPD who were treated with once-daily LABA/LAMA FDCs were included in the current study (Figure 1). Of the patients, 114 were treated with UMEC/VIL, 34 with IND/GLY, and 50 with TIO/OLO. Most of the patients were male (93.9%), with a mean age of 70.4 ± 10.4 years, mean weight of 64.2 ± 6.6 kg, and mean height of 163.1 ± 11.6 cm. The mean packs/year of smoking in our cohort was 40.2 ± 28.8. The most common comorbidity was congestive heart failure (CHF, *n* = 41, 20.7%), followed by coronary artery disease (CAD, *n* = 39, 19.7%). The mean FEV1/FVC ratio, FEV1%, FVC%, and RV% were 59.1 ± 9.9%, 55.7 ± 16.2%, 73.8 ± 16.5%, and 171.1 ± 63.1%, respectively. The GOLD III/IV group accounted for 36.2% (*n* = 71) of all study patients. The presence of comorbidities, the severity of GOLD stage, and the pulmonary function parameters (FEV1, FVC, and RV) before dual bronchodilator treatment showed no significant difference among the three examined LABA/LAMA FDCs (Table 1).

### 3.2. Pulmonary Function Changes Following Treatment

We evaluated whether there were any significant differences in the pulmonary function parameters before and after 12 months of treatment with the different medications (UMEC/VIL, IND/GLY, and TIO/OLO). Patients treated with UMEC/VIL displayed a significant improvement in the absolute FEV1 values (Δ = 0.099 L, *p* = 0.012). However, no significant difference was noted between the IND/GLY group and the TIO/OLO group (Δ = 0.116 L, *p* = 0.057; Δ = 0.056 L, *p* = 0.107) (Figure 2). Those treated with UMEC/VIL also displayed a significant improvement in absolute FVC values (Δ = 0.115 L, *p* = 0.031). However, those treated with IND/GLY and TIO/OLO did not display a significant improvement (Δ = 0.052 L, *p* = 0.532 and Δ = 0.118 L, *p* = 0.093) (Figure 3).

In addition, the percentage changes in pulmonary function parameters after 12 months of treatment were as follows. Patients treated with UMEC/VIL displayed a significant increase in the percentage changes in FEV1 (55.2% to 60.9%; Δ = 5.74%, *p* = 0.012) and FVC values (75.1% to 81.1%; Δ = 6.45%, *p* < 0.001) (Figure 2 and Figure 3). However, the patients treated with IND/GLY only displayed a significant increase in FEV1 values (58.2% to 63.6%; Δ = 5.37%, *p* = 0.023) (Figure 2). In the TIO/OLD users, all pulmonary function parameters significantly improved, including FEV1 (54.1% to 57.7%; Δ = 3.61%, *p* = 0.009), FVC (71.7% to 77.9%; Δ = 6.16%, *p* = 0.009), and RV (180.1% to 152.5%; Δ = −27.55%, *p* = 0.007) (Figure 2, Figure 3 and Figure 4). We found that TIO/OLO treatment might have more effects on lung hyperinflation due to the significant reduction in RV.

### 3.3. Minimal Clinically Important Differences in FEV1, FVC, and CAT Score Following Treatment

We also assessed lung function improvements and quality of life according to minimal clinically important differences (MCIDs) in FEV1, FVC, and CAT score among the three groups. Around 40% of the patients in the three groups reached an MCID in FEV1 (43.5% in UMEC/VIL, 41.2% in IND/GLY, 38.3% TIO/OLO). Patients treated with UMEC/VIL and TIO/OLO had a higher MCID in FVC (48.1%, 46.8%) than those treated with IND/GLY (33.3%). There was no significant difference in CAT score improvement among the three groups (Table 2).

## 4. Discussion

This was a real-world study to determine if treatments produced similar benefits in patients with COPD who were treated with different once-daily LABA/LAMA FDCs (UMEC/VIL, GLY/IND, and TIO/OLO), including an assessment of multiple pulmonary function parameters. The current study reported that pulmonary function improvement was significantly affected by the different types of medication. There was a statistically significant increase in FEV1 and FVC for patients treated with UMEC/VIL for 12 months. Patients treated with GLY/IND for 12 months only displayed a significant improvement in FEV1. However, all pulmonary function parameters (FEV1, FVC, and RV) were significantly improved in patients who received 12 months of TIO/OLO treatment; there was a particularly notable reduction in RV.

A meta-analysis of dual bronchodilation with LAMA/LABA in COPD patients, which enrolled fourteen papers and one congress abstract, and included 23,168 patients, found that all dual bronchodilation therapies elicited a greater increase in FEV1 than monotherapy alone. The IND/GLY combination increased FEV1 by 89.44 mL (95% CI, 76.04–102.85); FEV1 was increased by 54.75 mL following TIO/OLO combination treatment (95% CI, 45.70–63.80); UMEC/VIL combination treatment increased FEV1 by 83.66 mL (95% CI, 65.65–101.67) [9]. Another network meta-analysis of data extracted from 22 studies was performed by linking the efficacy and safety outcomes, and it reported that TIO/OLO > UMEC/VIL > GLY/IND following an indirect comparison [23]. However, published data regarding a direct comparison of TIO/OLO, UMEC/VIL, and GLY/IND are limited.

A direct comparison study indicated that at 8 weeks, UMEC/VIL was superior to TIO/OLO for the primary endpoint of FEV1 (167 mL in UMEC/VI and 110 mL in TIO/OLO, *p* = 0.001.) [12]. Another 12-week direct comparison study demonstrated that IND/GLY and UMEC/VI provided clinically meaningful and comparable bronchodilation [14]. In the current study, medication delivered via a DPI (IND/GLY and UMEC/VIL) resulted in a no significantly greater improvement in FEV1 compared with SMIs (TIO/OLO) after 52 weeks of treatment, which was consistent with previous reports. However, a real-world study from Taiwan reported that TIO/OLO was superior to IND/GLY and UMEC/VIL in relation to FEV1 (98.7 vs. 65.2 vs. 64.4 mL, respectively; *p* < 0.001) and FVC (127.3 vs. 58.2 vs. 79.1 mL; *p* < 0.001) after 12 months of treatment [15]. A prospective direct comparison study (open-labeled) in Japan of the three once-daily LABA/LAMA FDCs reported no significant differences in spirometry parameters after 12 weeks of treatment [16]. These few studies appear to be contradictory in their findings; however, this could be explained by the different study designs, the different patient cohorts, and the differences between retrospective real-world studies and RCTs. A more direct comparison of these three medications using RCTs might be needed in the future. Poor inhaler adherence and technique error can lead to poor COPD control. Lower adherence to medication in COPD was shown to be associated with a lower FEV1 and a higher risk of admission to hospital and death [24]. It is common that patients do not use their devices properly in real-world practice [25], and this can be affected by several factors, including the frequency of administration, device preference, and the patients’ understanding of COPD and the necessary treatment [19,25]. One study reported that fewer COPD patients made critical errors with ELLIPTA compared with other commonly used devices, and that most patients (57–70%) made no errors using ELLIPTA [26]. Our study reported that UMEC/VIL (delivered via ELLIPTA DPI) showed significant improvement in both FEV1 and FVC. The increased convenience of using the ELLIPTA DPI may be one of the influencing factors.

Small airway disease (SAD) is a key pathological feature of COPD. Patients with COPD accompanied by SAD have worse respiratory reactance, worse spirometry results, more severe lung hyperinflation, and a poorer health status [27]. Most inhaled therapies cannot deliver to small airways sufficiently [28]. An in vitro study tried to compare drug delivery to the lung among four different inhalers (Respimat, Breezhaler, Genuair, and ELLIPTA). The modeled dose to the lung was found to be highest in Respimat (59% for moderate COPD; 67% for severe COPD). Respimat showed the lowest number of particles depositing in the mouth–throat model and the highest number reaching all regions of the simulation lung model [29]. The measurement of FEV1 via spirometry is not specific to SAD, with larger airways contributing substantially to the expired volume [30]. COPD patients with small airway involvement usually have more gas trapping with a higher RV [31,32]. SAD was also associated with bronchodilator responsiveness in terms of FVC, but not in terms of FEV1 [27]. Our study reported that TIO/OLO (delivered via Respimat SMI) tended to show an increased improvement in FVC and RV among the three different medications, indicating an improvement in SAD, because the lung deposition would be better with Respimat compared with the DPI therapies [29]. Cheng et al. also reported that patients treated with TIO/OLO had significantly better improvements in FEV1 and FVC compared to those treated with UMEC/VIL or IND/GLY in COPD patients with low FEV1 or FVC, which means more hyperinflation [29]. We need more real-world studies rather than in vitro studies to confirm this concept.

Airflow limitation, as diagnosed via spirometry, remains the standard diagnosis for COPD. Most previous studies have relied on FEV1 or FVC alone for an assessment of disease severity or the reversibility of bronchodilators [33,34,35,36]. However, a measurement of FEV1 alone is not sufficient to determine the complexity of COPD, especially in those with SAD. Lung volume, including inspiratory capacity (IC) or RV, are potentially useful parameters for detecting the response to bronchodilator therapy [37,38]. The additional measurement of FVC in the present study could help detect more bronchodilator responders compared with FEV1 alone [36,39]. The current study aimed to provide a comprehensive overview of the common parameters (FEV1, FVC, and RV) of spirometry in our clinical practice, to help distinguish the different effects of these three medications. COPD patients treated with TIO/OLO showed improvement in all measured pulmonary function parameters, especially in RV. After 52 weeks of treatment, IND/GLY and UMEC/VIL showed more improvement in FEV1 and FVC as opposed to RV in COPD patients. We conducted this study to provide additional data to primary care clinical physicians to help enhance the efficacy of COPD management.

There were several limitations in the current study. First, this was a retrospective observational study, which was performed at a single medical center with a relatively small number of patients and some degree of selection bias. Following the inclusion criteria, people who did not use the same once-daily LABA/LAMA dual bronchodilator FDC continuously for 12 months were excluded. The decision regarding which type of medication to treat patients with was made by their primary care physician. The treatment strategy is based on the clinical physician’s judgement, the patients’ preference and adherence, and the medication available. UMEC/VIL was the first available once-daily LABA/LAMA FDC in our hospital; therefore, the number of patients in each group was imbalanced. We tried to use two parameters to evaluate the difference before and after the LABA/LAMA FDC therapy, including the absolute value (FEV1, FVC) and the predict value in each group (FEV1% and FVC%), which showed inconsistent results in these two values. However, FEV1% and FVC% were possibly more accurate than the absolute values of FEV1 and FVC, which were not influenced by age, sex, body weight, and body height. Therefore, the results of this study were based on the predicted values of FEV1%, FVC%, and RV%. Third, we might need more parameters of spirometry for a full SAD evaluation, including IC, total lung capacity (TLC), and RV/TLC. We chose the three most commonly used parameters of spirometry, and RV can be used to replace IC in our clinical practice. Newton et al. reported a significant reduction in RV and increases in FVC and IC following the administration of a bronchodilator in moderate to severe hyperinflated COPD patients [37]. Fourth, we did not divide patients with COPD into those with emphysema and chronic bronchitis. The phenotype may influence the treatment results. Fifth, the CAT values were relatively low in the present study. The possible reasons included: 1. The CAT scores were collected from medical charts at the beginning of treatment for those treated with LABA/LAMA FDCs. They may have been treated with other therapy before inclusion, such as monotherapy or ICS/LABA. 2. Asian people tend to be more conservative in reporting their physical conditions. Another multicenter study from Taiwan reported similar results [40]. Finally, all baseline data were available, but some follow-up data were missing. There were few differences in baseline lung function data before and after performing the paired *t*-test. The baseline pulmonary function parameters (FEV1, FVC, and RV) still showed no significant difference among the three LABA/LAMA FDCs after performing the paired *t*-test (Appendix A). Despite these limitations, the current study provides the first real-world comparison of the efficacy of the three most commonly used once-daily LABA/LAMA FDCs, and it was conducted via comprehensive measurements of spirometry parameters.

## 5. Conclusions

Patients with COPD treated with either of the three once-daily LABA/LAMA FDCs for 12 months showed improvements in pulmonary lung function, including FEV1, FVC, and RV. All of the patients treated with once-daily FDCs had improvements in FEV1. The UMEC/VIL group also had significant improvements in FVC. The TIO/OLO (agents delivered by SMI) group not only displayed a significant improvement in FVC but displayed a reduction in RV, which may indicate that it is beneficial for improving small airway obstruction.

## Figures and Tables

**Figure 1 jcm-11-07165-f001:**
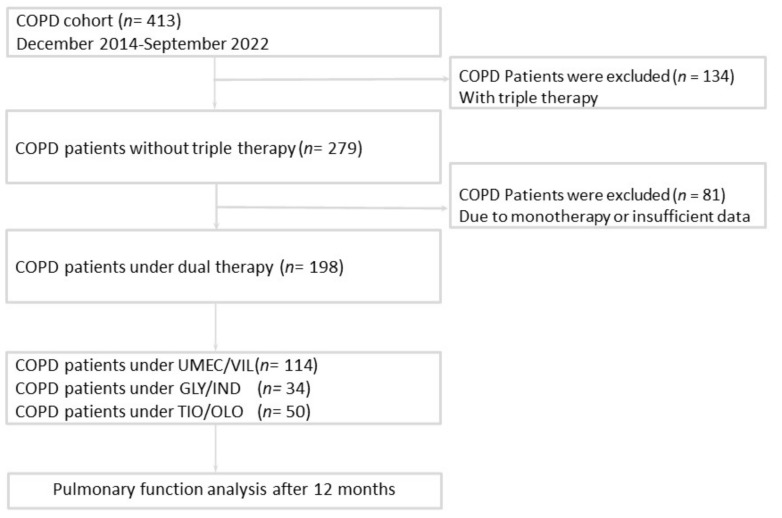
Patient enrollment flowchart. COPD: chronic obstructive pulmonary disease; UMEC/VIL: umeclidinium/vilanterol; GLY/IND: glycopyrronium/indacaterol; TIO/OLO: tiotropium/olodaterol.

**Figure 2 jcm-11-07165-f002:**
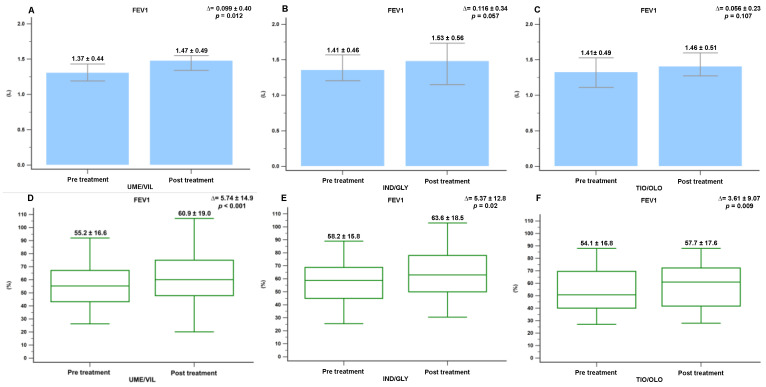
The significant difference in FEV1(L) and FEV1(%) in COPD patients before and after 12 months of treatment with medications (**A**) FEV1(L) in UMEC/VIL, (**B**) FEV1(L) in IND/GLY, (**C**) FEV1(L) in TIO/OLO, (**D**) FEV1(%) in UMEC/VIL, (**E**) FEV1(%) in IND/GLY, and (**F**) FEV1(%) in TIO/OLO. Data were presented as mean and standard deviation.

**Figure 3 jcm-11-07165-f003:**
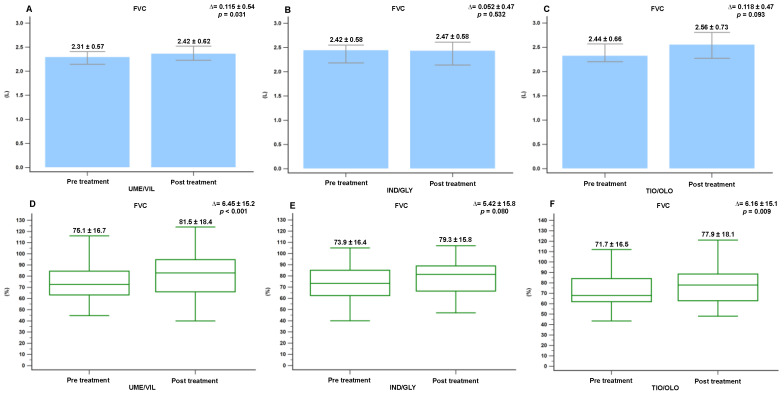
The significant difference in FVC(L) and FVC (%) in COPD patients before and after 12 months of treatment with medications (**A**) FVC(L) in UMEC/VIL, (**B**) FVC(L) in IND/GLY, (**C**) FVC(L) in TIO/OLO, (**D**) FVC(%) in UMEC/VIL, (**E**) FVC(%) in IND/GLY, and (**F**) FVC(%) in TIO/OLO. Data were presented as mean and standard deviation.

**Figure 4 jcm-11-07165-f004:**
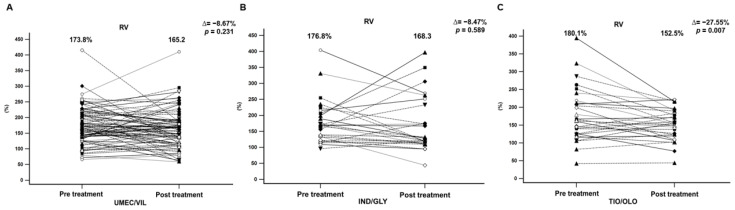
The significant difference in RV (%) in COPD patients before and after 12 months of treatment with (**A**) UMEC/VIL, (**B**) IND/GLY, and (**C**) TIO/OLO. Data were presented as mean and standard deviation.

**Table 1 jcm-11-07165-t001:** Patient characteristics.

	Total (*n* = 198)	UMEC/VIL (*n* = 114)	IND/GLY (*n* = 34)	TIO/OLO (*n* = 50)	*p*-Value
Age, years (SD)	70.4 (10.4)	70.6 (10.9)	72.4 (7.9)	68.5 (10.3)	0.229
Male (%)	186 (93.9)	109 (95.6)	32 (94.1)	45 (90.0)	0.381
Body Height, cm (SD)	163.1 (6.6)	163.1 (7.0)	162.3 (6.9)	163.6 (5.7)	0.651
Body Weight, kg (SD)	64.2 (11.6)	63.9 (11.3)	65.5 (11.7)	63.7 (12.2)	0.74
BMI, kg/m^2^ (SD)	24.2 (4.1)	24.1 (4.1)	24.8 (3.8)	23.7 (4.3)	0.483
Smoking, packs/year (SD)	40.2 (28.8)	40.3 (29.2)	48.2 (31.5)	34.5 (25.1)	0.100
ACO (%)	14 (7.1)	7 (6.1)	4 (11.8)	3 (6.0)	0.502
CAD (%)	39 (19.7)	22 (19.3)	7 (20.6)	10 (20.0)	0.984
CHF (%)	41 (20.7)	24 (21.1)	6 (17.6)	11 (22.0)	0.881
Cancers (%)	39 (19.7)	22 (19.3)	8 (23.5)	9 (18.0)	0.811
Bronchiectasis (%)	15 (7.6)	8 (7.0)	3 (8.8)	4 (8.0)	0.932
FEV1/FVC, % (SD)	59.1 (9.9)	59.4 (9.7)	59.3 (8.1)	58.1 (11.3)	0.74
FEV1, L (SD)	1.38 (0.45)	1.36 (0.43)	1.41 (0.46)	1.41 (0.48)	0.788
FEV1, % (SD)	55.7 (16.2)	55.3 (16.3)	58.2 (15.7)	54.8 (16.7)	0.610
FVC, L (SD)	2.35 (0.59)	2.29 (0.56)	2.38 (0.61)	2.46 (0.65)	0.214
FVC, % (SD)	73.8 (16.5)	74.8 (17.1)	73.9 (16.3)	72.3 (15.9)	0.713
RV, % (SD)	171.1 (63.1)	170.3 (57.5)	173.4 (68.5)	171.2 (74.1)	0.972
MMEF25-75, % (SD)	27.9 (12.1)	29.1 (11.2)	27.2 (9.7)	26.5 (14.8)	0.551
GOLD stage, %					0.448
I	15 (7.7)	9 (8.0)	2 (5.9)	4 (8.0)	
II	110 (56.1)	65 (58.0)	23 (67.6)	22 (44.0)	
III	61 (31.1)	33 (29.5)	7 (20.6)	21 (42.0)	
IV	10 (5.1)	5 (4.5)	2 (5.9)	3 (6.0)	
CAT (IQR)	6.5 (6.5–7)	6.0 (4–8.75)	6.5 (4–10.0)	7.5 (5.0–13.0)	0.058
AE in last year (IQR)	0 (0–0)	0 (0–1)	0 (0–1)	0 (0–0)	0.270

ACO: asthma and COPD overlap; AE: acute exacerbation; BMI: body mass index; CAD: coronary heart disease; CAT: COPD Assessment Test; CHF: congestive heart failure; FEV1: forced expiratory volume in 1 s; FVC: forced vital capacity; GOLD: the Global Initiative for Chronic Obstructive Lung Disease; IND/GLY: Indacaterol/Glycopyrronium; MMEF: Maximal Mid-Expiratory Flow; IQR: interquartile range; RV: residual volume; SD: standard deviation; TIO/OLO: tiotropium/olodaterol; UMEC/VIL: umeclidinium/vilanterol.

**Table 2 jcm-11-07165-t002:** The MCID of FEV1, FVC, and CAT after these three medicines treatment.

	UMEC/VIL (*n* = 114)	IND/GLY (*n* = 34)	TIO/OLO (*n* = 50)	*p*-Value
FEV1, MCID (%)	49 (43.5)	14 (41.2)	19 (38.3)	0.868
FVC, MCID (%)	55 (48.1)	11 (33.3)	27 (46.8)	0.316
CAT, MCID (%)	30 (28.0)	10 (30.3)	14 (28.5)	0.873

CAT: The COPD Assessment Test; FEV1: forced expiratory volume in one second; FVC: forced vital capacity; IND/GLY: Indacaterol/Glycopyrronium; MCID: minimal clinically important difference; TIO/OLO: tiotropium/olodaterol; UMEC/VIL: umeclidinium/vilanterol.

## Data Availability

All data generated or analyzed during this study are included in this published article.

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
