# Peer review of "Differences in Pulmonary Function Improvement after Once-Daily LABA/LAMA Fixed-Dose Combinations in Patients with COPD"

_jcm, 2022, doi:10.3390/jcm11237165_

Round 1

Reviewer 1 Report

There are some significant problems with the presented data. It is very peculiar that the groups match so well regarding all the pre-treatment characteristics and that there are no significant differences found even though the study is not randomized. This looks like authors peeked patients up and that the study is based not based on a random sample. The methods are not described at all and especially not the lung function measurements that are the key (main outcome) of the study. We don't know how RV was measured. The CAT values are extremely low for such a sample of very old male group of subjects suggesting no respiratory symptoms, so the diagnosis of COPD is doubtful. Also, the conclusions are not based on presented data or from the presented data this cannot be seen. Figures are not properly marked in the legends. Not all significant limitations are presented or discussed in the Discussion under limitations.

Reviewer 2 Report

The paper evaluates the long-time administration (12 months) of fixed dose combinations of long-acting β2-agonist (LABA)/long-acting muscarinic antagonist (LAMA) in patients with chronic obstructive pulmonary disease (COPD). The effectiveness of the treatment was evaluated by pulmonary functional tests (FEV1, FVC and RV measurement). It is a real-word retrospective study that includes data from COPD patients treated over 5 years in a single hospital unit. The study compares three therapeutic variants of LABA/LAMA FDC. After applying the inclusion and exclusion criteria, the study enrolled 198 patients of which 114 were treated with UMEC/VIL, 34 with IND/GLY, and 50 with TIO/OLO.

The study design used is appropriate to test the hypothesis and is properly described in text by using a suggestive diagram for the steps followed during patient monitoring. The data interpretation is explained in a very clear manner using graphical representations of data statistical analyses. Before starting the data collection and analysis, the authors obtained ethical approval of the Hospital review Board but he requirement for informed patient consent was waived because of the study’s retrospective design. The results obtained are discussed comparatively with the bibliographic references and the conclusions are consistent with the evidence and arguments presented.  

However, a complete revision of the text formatting is necessary, because there appear paragraphs or parts of sentences written with different characters type and size.

Reviewer 3 Report

Huang et al. describe a retrospective study designed to differentiate between the effectiveness of three mainstream LAMA/LABA fixed-dose combination therapy for COPD. The primary endpoint of the study measures FEV1, FEV1%, FVC, and RV by spirometry before the start of treatment and at the time 12 months of treatment. The authors concluded that all three combination treatments resulted in an improvement of lung function (FEV1 and FEV1%), although not all were significant. They also concluded that UMEC/VIL and TIO/OLO combination treatments were superior in increasing the forced expiratory volume compared to IND/GLY. In addition, TIO/OLO combination presented a reduction in residual volume.

Critics:

1)      Besides the spirometry data, the statistical analysis is the most crucial part of this study. The statistical analysis in methodology is not detailed enough and does not include any precise description of which data were analyzed and presented by which statistical analysis. Also, in the body of the text, the mean value should be presented as mean±SD as it applies.

2)      Please correct the phrase “improving spirometry in patient …” (lines 34-35). The goal of the treatment is to improve the lung function of patients with COPD, not improve spirometry results.

3)       Also please correct the sentence in lines (35-36) “Patients with 35 COPD who were treated with once-daily LABA/LAMA FDCs for 12 months were enrolled.” In a retrospective study, patients can not be enrolled; only the measured data from patients can be included.

4)      In line 41, Forced expiratory volume in 1 second is (FEV1). The FEV1% represents the FEV1/FVC ratio. Please correct it.

5)      Also please define FVC% and RV% because these values are also calculated and not measured.

6)      Patients` FEV1 values in IND/GLY and TIO/OLO groups represent a higher deviation at the time of pretreatment compared to UMEC/VIL which may result in an increased deviation after treatment. So, the significance and superiority of UMEC/VIL treatment may originate from the non-equally distributed patient population between the treatment groups. The FEV1% significantly improved in all 3 treatments based on their p values (<0.001, =0.023, =0.009). All three values are below the 0.05 significance limit. How could it be explained that the FEV1% values are significant in all three treatment groups derived from the FEV1 and FVC absolute values if the absolute values are not significant between the pretreatment and post-treatment groups regarding IND/GLY or TIO/OLO treatment?

7)      The same unequal distribution of patients between the three treatment groups may explain the significant FVC improvement in the case of UMEC/VIL and the non-significant change in the case of the other two treatment combinations.

8)      I would suggest analyzing the data from the perspective of COPD with emphysema or chronic bronchitis standpoints. Patients with extensive emphysema would have a larger residual volume that can not be significantly improved with any treatment. Contrary, patients with chronic bronchitis and no or minimal emphysema would have trapped air as an increased RV that would improve with bronchodilator treatment. 

Reviewer 4 Report

INTRODUCTION The aims of the study need to be clarified. As the allocation to the different treatments is not randomized, it is not possible to make a comparison between the three treatments as there could easily be biases beyond the drugs themselves that favour one treatment over the others. The  purpose of "real world" studies is to see if the treatments produce similar benefits in regular clinic patients without close supervision, as in clinical trial patients.

The physiological measurements provide insights into the mechanisms of how the drugs work, but they do not provide evidence of one treatment being superior to others in a study that is not randomized.

It should be stated whether spirometry measurements are performed before or after the daily dose. Even though the drugs are inhaled once daily, variations in their half lives may be responsible for differences in trough measurements.

METHODS Analysis of data obviously has to be retrospective (data cannot be analysed before it has been collected). The important point is the timing of data collection. Were the data collected as the patients had their reviews during the course of their clinical care, or did some-one have to go back through files etc to find what data they could? The former would be prospective, and the latter would be retrospective.

It is not clear if all the patients in the study were commenced on LAMA / LABA at the start of the twelve month period, or if they had been on it for some time before the twelve month data collection period commenced. If they had been on LAMA / LABA previously we would need to know for how long.

RESULTS

Did patients swap between different LAMA / LABA inhalers during the twelve months?

Over the six years of the study did the pattern of prescribing different LAMA / LABAs change?

The CAT score seem rather low in Table 1. Is there an explanation for this?

DISCUSSION Did any changes in any of the physiological parameters predict changes in CAT scores?

How do the changes in these patients compare with changes that have been published in the definitive clinical trials?

CONFLICTS OF INTERESTS As there are none declared, does this mean that none of their authors or their departments receive funding from respiratory pharmaceutical companies for this or other studies?

Round 2

Reviewer 1 Report

Although some improvements have been made this is still looks like a significantly biased analysis. In your cover letter you mentioned a comparable published study (Hsieh M-J, Chen N-H, Cheng S-L, Tao C-W, Wei Y-F, Wu Y-K, et al. Comparing Clinical Outcomes of Tiotropium/Olodaterol, Umeclidinium/Vilanterol, and Indacaterol/Glycopyrronium Fixed-Dose Combination Therapy in Patients with Chronic Obstructive Pulmonary Disease in Taiwan: A Multicenter Cohort Study. International Journal of Chronic Obstructive Pulmonary Disease. 2022;17:967. ) that you haven't included in your manuscript and you still have a comment in your discussion: "To the best of our knowledge, this is the first study to see if the treatments produce 244 similar benefits in patients with COPD who were treated with different once-daily 245 LABA/LAMA FDCs (UMEC/VIL, GLY/IND, and TIO/OLO), including an assessment of 246 multiple pulmonary function parameters." 

There are still flaws regarding the statistical analysis that should use repeated measures ANOVA analysis to compare changes between different treatments. Also the figures are still not properly marked (your have graphs A-F on your figures and in the legends you only mention A-C). Also there is no legend about what the presented values stand for (mean, median, SDs, etc.). Figure 4 lacks central values and variance markings.

Also you can't overcome the problem of imbalanced group samples with what you mention: "To overcome this problem, we used these two parameter values (FEV1 and FEV1%; FVC and FVC%) in each group to evaluate the difference before and after the 337 LABA/LAMA FDC therapy, which showed inconsistent results in these two values."

In the Discussion you also discuss point that you haven't investigate in your study which is purely speculative and not scientific, like: "Our study reported 288 that UMEC/VIL (delivered via ELLIPTA DPI) showed the greatest improvement in FEV1 and FVC among the three different medications, however, this could be associated with the increased convenience of using the ELLIPTA DPI."

Reviewer 3 Report

The authors revised the content of the manuscript according to the suggestions. The present version provides a more accurate presentation of the statistical analysis. The study's limitations originate from the retrospective design (such as unbalanced patient numbers between patients` groups), which is discussed by the authors at the end of the manuscript.  

Minor grammatical changes:

·        In this version of the manuscript  Figure 1 flowchart contains extra characters. Please clean up this figure.

·        In line 311: the “rather than” phrase is still duplicated. Please delete one of them.
